# Bioinformatics Analyses Identify the Therapeutic Potential of ST8SIA6 for Colon Cancer

**DOI:** 10.3390/jpm12030401

**Published:** 2022-03-04

**Authors:** Chou-Yuan Ko, Tian-Huei Chu, Ching-Cheng Hsu, Hsin-Pao Chen, Shih-Chung Huang, Chen-Lin Chang, Shiow-Jyu Tzou, Tung-Yuan Chen, Chia-Chen Lin, Pei-Chun Shih, Chung-Hsien Lin, Chuan-Fa Chang, Yung-Kuo Lee

**Affiliations:** 1Division of Gastroenterology and Hepatology, Department of Internal Medicine, Kaohsiung Armed Forces General Hospital, Kaohsiung 80284, Taiwan; gastroenterokjy@gmail.com; 2Institute of Medical Science and Technology, National Sun Yat-sen University, Kaohsiung 80424, Taiwan; sghung@gmail.com (S.-C.H.); changchenling@gmail.com (C.-L.C.); jyu0120@gmail.com (S.-J.T.); 3Medical Laboratory, Medical Education and Research Center, Kaohsiung Armed Forces General Hospital, Kaohsiung 80284, Taiwan; skbboyz0817@gmail.com; 4Department of Internal Medicine, Division of Cardiology, The University of Texas Southwestern Medical Center, Dallas, TX 75390, USA; ching-cheng.hsu@utsouthwestern.edu; 5Institute of Basic Medical Science, College of Medicine, National Cheng Kung University, Tainan 70101, Taiwan; jdgek54690@hotmail.com (P.-C.S.); joey2719@gmail.com (C.-H.L.); 6Department of Surgery, E-DA Hospital, I-Shou University, Kaohsiung 82445, Taiwan; shinpao2002@yahoo.com.tw; 7Division of Cardiology, Department of Medicine, Kaohsiung Armed Forces General Hospital, Kaohsiung 80284, Taiwan; 8Department of Psychiatry, Kaohsiung Armed Forces General Hospital, Kaohsiung 80284, Taiwan; 9Department of Nursing, Kaohsiung Armed Forces General Hospital, Kaohsiung 80284, Taiwan; 10Division of Colorectal Surgery, Department of Surgery, Kaohsiung Armed Forces General Hospital, Kaohsiung 80284, Taiwan; sur045@yahoo.com.tw; 11Clinical Pathology Department, Chi Mei Medical Center, Tainan 71004, Taiwan; cchia1028@gmail.com; 12Department of Medical Laboratory Science and Biotechnology, College of Medicine, National Cheng Kung University, Tainan 70101, Taiwan

**Keywords:** bioinformatics application, colon adenocarcinoma (COAD), ST8 sialyltransferases 6, biological network, tumor immunity, immunotherapy responses

## Abstract

Sialylation of glycoproteins is modified by distinct sialyltransferases such as ST3Gal, ST6Gal, ST6GalNAc, or ST8SIA with α2,3-, α2,6-, or α2,8-linkages. Alteration of these sialyltransferases causing aberrant sialylation is associated with the progression of colon cancer. However, among the ST8- sialyltransferases, the role of ST8SIA6 in colon cancer remains poorly understood. In this study, we explored the involvement of ST8SIA6 in colon cancer using multiple gene databases. The relationship between ST8SIA6 expression and tumor stages/grades was investigated by UALCAN analysis, and Kaplan–Meier Plotter analysis was used to analyze the expression of ST8SIA6 on the survival outcome of colon cancer patients. Moreover, the biological functions of ST8SIA6 in colon cancer were explored using LinkedOmics and cancer cell metabolism gene DB. Finally, TIMER and TISMO analyses were used to delineate ST8SIA6 levels in tumor immunity and immunotherapy responses, respectively. ST8SIA6 downregulation was associated with an advanced stage and poorly differentiated grade; however, ST8SIA6 expression did not affect the survival outcomes in patients with colon cancer. Gene ontology analysis suggested that ST8SIA6 participates in cell surface adhesion, angiogenesis, and membrane vesicle trafficking. In addition, ST8SIA6 levels affected immunocyte infiltration and immunotherapy responses in colon cancer. Collectively, these results suggest that ST8SIA6 may serve as a novel therapeutic target towards personalized medicine for colon cancer.

## 1. Introduction

Colon cancer is the most common gastrointestinal malignancy. More than 2 million new colorectal cancer cases and 900,000 deaths have been reported in GLOBOCAN 2020 [1]. Primary prevention and cancer screening remain key strategies in colon cancer treatment [2]. Therefore, it is necessary to identify potential diagnostic biomarkers for colon cancer at an early stage.

Several studies have shown that protein or lipid sialylation plays a crucial role in post-translational modifications during malignant tumor progression [3,4]. Most sialyltransferases are responsible for adding single sialic acid to glycan chains via α2,3-, and α2,6-, respectively. Terminal sialic acids play an important role in controlling cellular activation and transformation differentiation by regulating cellular interactions with ligands and neighboring cells [5]. Sialyl Lewis x, sialyl Lewis a, sialyl Tn, and polysialic acid are well-studied and common carbohydrate tumor markers for the diagnosis, prognosis, and prediction of cancer [6,7,8,9,10]. Moreover, disialic, oligosialic, and polysialic acids can be obtained by the addition of sialic acids via α2,8-linkage [11]. α2,8-linkage sialylation is catalyzed by six membranes of the α2,8 sialyltransferases (ST8SIA) family, which has been correlated with clinical outcomes [12]. Polysialic acid on cell surface proteins is controlled by increasing ST8SIA2 and ST8SIA4 in metastatic neuroblastoma [13,14]. Disialic acid on glycoproteins in mammals is associated with ST8SIA3 and ST8SIA6 in the human brain and other tissues [15,16,17,18,19,20]. According to Takashima et al. and Teintenier-Lelievre et al. studies have revealed mouse or human ST8SIA6 generates disialic acid in *O*-glycosylproteins. ST8SIA6 generates disialic acids that have been found in the α2,3-sialylated core 1 structure (Neu5Acα2,8Neu5Acα2,3Galβ1,34GalNAc-*O*-Ser) [18,21] and is important for binding to human Siglec-7 and murine Siglec-E. The interaction between Siglec-7 and surface glycoproteins or glycolipids with disialylation has been associated with human disease progression [22,23,24]. In a murine model, ST8SIA6 and Siglec-E expression was involved in tumor progression and diabetes by altering macrophage polarization and inflammation status [25,26].

It has rarely been determined whether human disease related disialylation is catalyzed by ST8SIA6, and the mechanism of colon cancer progression remains still unknown. The ability to analyze the levels of ST8SIA6 in histological and biological samples, and in cellular interaction networks and immunotherapy responses, is required to develop effective approaches for medical intervention. In this study, systematic bioinformatics analysis of global web resources was highly useful for exploring the function of ST8SIA6 in transcriptional alterations, functional networks, and tumor immunity. Our results could reveal new targets for colon cancer screening. Furthermore, it is also important to investigate potential prognostic markers for the tracing and improvement of patient outcomes after treatment in patients with advanced-stage colon cancer.

## 2. Materials and Methods

### 2.1. UALCAN Database Analysis

UALCAN (http://ualcan.path.uab.edu/, accessed on 24 February 2022) was used to reveal the relative expression of genes across normal and tumor samples with various subgroups based on individual cancer stages, tumor grade, or other clinicopathological features according to The Cancer Genome Atlas (TCGA) RNA-seq data from 31 cancer types [27].

### 2.2. Kaplan–Meier Survival Curve Analysis

Kaplan–Meier Plotter (http://kmplot.com/analysis, accessed on 24 February 2022) was used to assess the effect of genes on survival rates in different cancer types [28]. The correlation between ST8SIA6 expression and overall survival (OS) or relapse-free survival (RFS) of colon cancer were analyzed using Kaplan–Meier Plotter.

### 2.3. LinkedOmics Database Analysis

The LinkedOmics database (http://www.linkedomics.org/admin.php, accessed on 24 February 2022) is a web-based platform for analyzing cancer-associated multi-omics data from 32 TCGA [29]. ST8SIA6 co-expressed genes were analyzed statistically using Pearson’s correlation coefficient and are presented as heat maps. Gene set enrichment analysis (GSEA) enriched the functional networks associated with ST8SIA6, including Gene Ontology (GO) and Kyoto Encyclopedia of Genes and Genomes (KEGG) pathways. The rank criterion was an FDR < 0.05.

### 2.4. Cancer Cell Metabolism Gene DataBase Analysis

A cancer cell metabolism gene database (http://bioinfo.mc.vanderbilt.edu/ccmGDB, accessed on 24 February 2022) was used to identify the genes involved in cancer metabolism [30]. Interaction networks used the top 20 co-expressed genes for ST8SIA6. ST8SIA6 is colored in red and metabolism pathways associated with ST8SIA6 are in orange.

### 2.5. TIMER2.0 Database Analysis

The TIMER2.0 database (http://timer.cistrome.org, accessed on 24 February 2022) is a web server upgraded from the Tumor Immune Estimation Resource (TIMER) for the systematic analysis of six major tumor-infiltrating immune subsets from 32 cancer types [31,32]. The association between tumor-infiltrating immune cells and ST8SIA6 was explored. The module of somatic copy number alteration (SCNA) provided each immune cell subset with the somatic copy number aberrations of ST8SIA6 in colon cancer.

### 2.6. TISMO Database Analysis

Tumor Immune Syngeneic MOuse (TISMO) database (http://tismo.cistrome.org, accessed on 24 February 2022) is a resource of syngeneic mouse models and tumor models for the analysis of tumor immunity and immunotherapy response in 23 cancer types [33]. Interactions with colon cancer and immune regulation were studied according to cytokine treatment and immune checkpoint blockade (ICB) response in syngeneic colon cancer cell lines with ST8SIA6 gene expression.

### 2.7. Statistical Analysis

A *t*-test *p* < 0.05 was used to show the significance between the subgroups with the gene expression levels of ST8SIA6. Kaplan–Meier curves were used to compare the survival times of the different groups. The log-rank test *p* < 0.05 was used to indicate the significance of the survival time.

## 3. Results

### 3.1. Low ST8SIA6 Expression Promotes Colon Cancer Progression

To investigate the role of ST8SIA6 in colon cancer patients, we evaluated the RNA transcription levels of ST8SIA6 in multiple colon cancer studies from TCGA. The mRNA expression of ST8SIA6 from the UALCAN database was significantly decreased in colon adenocarcinoma (COAD) tissues compared to that in normal tissues (Figure 1A). We further analyzed subgroups based on histological subtypes, disease stages, and nodal metastasis status of TCGA colon cancer samples in the UALCAN database, which showed lower transcriptional levels of ST8SIA6 in colon cancer patients with high-grade patterns than in normal controls (Figure 1B–D). Moreover, data from the subgroup analysis showed different transcription levels of ST8SIA6 based on race and body weight (Figure 1E,F). Thus, ST8SIA6 may serve as a potential diagnostic marker for high-grade colon tumors.

### 3.2. ST8SIA6 Expression Is Not Associated with Survival Outcomes in Colon Cancer

To determine the relationship between ST8SIA6 expression and survival rate in colon cancer, Kaplan–Meier survival curves were constructed to assess the survival outcomes. Survival information from the Kaplan–Meier plotter database in colon cancer cohorts indicated an association between ST8SIA6 mRNA expression and survival outcomes. The two groups were separated from colon cancer patients according to the median ST8SIA6 expression level in each cohort. The collected data in the colon cancer cohort showed that overall survival (OS; log-rank test, *p* = 0.096) (Figure 2A) and relapse-free survival (log-rank test, *p* = 0.14) (Figure 2B) had no significant difference in the high ST8SIA6 expression group and the low ST8SIA6 expression group.

### 3.3. Enriched Co-Expression Genes Networks with ST8SIA6 in Colon Cancer

To explore the biological functions of ST8SIA6 in the colon cancer database, the co-expression genes with ST8SIA6 from the LinkedOmics were used to examine. As shown in Figure 3A, red dots were significant positive correlations with ST8SIA6, whereas green dots were significant negative correlations (false discovery rate, FDR < 0.01). The top 50 significant genes positively correlated with ST8SIA6 are shown in the heat map (Figure 3B). The data showed that ST8SIA6 gene expression has a strong positive correlation with expression of TDP2 (r = 0.3907, *p* = 2.8714 × 10^−15^), PLOD2 (r = 0.3722, *p* = 6.71 × 10^−14^), FREM1 (r = 0.3507, *p* = 2.06 × 10^−12^), ARHGAP18 (r = 0.3472, *p* = 3.527 × 10^−12^), and FABP1 (r = 0.3455, *p* = 4.598 × 10^−12^), etc. (Figure 4A–E). Gene–gene interaction networks of ST8SIA6 and the top five co-expressed genes from the Cancer Cell Metabolism Gene Database (ccmGDB) are shown in Figure 4F. Thus, it is essential to understand the crosstalk between co-expressed genes and ST8SIA6 regulation in metabolic pathways. Gene Ontology (GO) term annotation by gene set enrichment analysis (GSEA) showed that genes co-expressed with ST8SIA6 participate in cell–cell or plasma membrane adhesion, angiogenesis, vesicle-mediated transport, amoeboid-type cell migration, and inositol lipid-mediated signaling (Figure 5A). The molecular function ontologies from GO annotation showed that ST8SIA6 co-expressed genes are involved in binding to the extracellular matrix and glycosaminoglycan (Figure 5B). Kyoto Encyclopedia of Genes and Genomes (KEGG) pathway-related Panther or Reactome analysis showed the enrichment pathways related to ST8SIA6 including integrin pathway, TGF-beta pathway, integrin cell surface interaction, extracellular matrix organization, molecules associated with elastic fibers, and cell surface interaction with the vascular wall, etc. (Figure 5C,D). These results suggest that the widespread impact of ST8SIA6 expression in colon cancer on the metastasis transcriptome has been analyzed in more detail.

### 3.4. ST8SIA6 Is Related to Immune Infiltration Level and Immunotherapy Response in Colon Cancer

Previous reports have shown that aberrant sialylation is related to immune modulation during cancer treatment [34,35]. To investigate whether ST8SIA6 expression is correlated with immune infiltration levels in colon cancer, we used the TIMER database. The results showed that ST8SIA6 expression has significant correlations with the infiltration levels of CD4^+^ T cells (r = 0.165, *p* = 9.19 × 10^−4^), macrophages (r = 0.136, *p* = 6.30 × 10^−3^), neutrophils (r = 0.1, *p* = 4.62 × 10^−2^), and dendritic cells (r = 0.108, *p* = 2.98 × 10^−2^) (Figure 6A). Moreover, somatic copy number alterations in ST8SIA6 had been significantly correlated with the infiltrating levels of B cells and CD8^+^ T cells (Figure 6B). In addition, we identified the expression of ST8SIA6 in cancer immunotherapy using TISMO database analysis. The results showed that ST8SIA6 expression was associated with different immunotherapeutic responses (Figure 7A,B). The data showed that after antiPD1 treatment, ST8SIA6 expression levels in the CT26 model are significantly upregulated in the antiPD1 responders, but not in the non-responders. Thus, after immunotherapy, the expression of ST8SIA6 is related to response for immune infiltration and the tumor microenvironment.

## 4. Discussion

Medical screening for colon cancer has been developed over the past 20 years [36]. Combined or single analysis assays of identified tumor markers, including carcinoembryonic antigen (CEA), carbohydrate antigen (CA 19.9), and tumor-associated glycoprotein-72 have been used to detect the transformation of cancer cells for colon cancer diagnosis [37,38,39]. These carbohydrate markers can directly detect glycosylation alternations for colon cancer diagnosis and treatment. Therefore, the development of a glycobiological approach has received much attention as a new diagnostic and targeted treatment method for colon cancer. These proteins with α-2,3, α-2,6, or α-2,8 linkages to sialylation are common oncogenic features of malignant tumors, which increase cancer cell migration and metastasis and remodel the tumor microenvironment (TME) to escape immune surveillance for targeted immunotherapy [40]. To obtain more detailed information on the potential gene functions of α-2,8 linkages to sialylation and regulatory interaction network with α-2,8 sialyltransferases, we performed a tumor marker of ST8SIA6 gene expression using high-throughput bioinformatics analysis from medical and biological data in colon cancer (Figure 8).

Initially, we demonstrated that ST8SIA6 mRNA levels were significantly decreased in colon cancer tissues compared to normal tissue by analyzing the transcriptome from clinical patient data that was categorized across various subgroups. In particular, obesity in colon cancer patients has been associated with ST8SIA6 expression [41]. In our study, we demonstrated the correlation between the downregulation of ST8SIA6 and clinicopathological characteristics or the risk of colon cancer from TCGA database. However, the results of KM plots revealed ST8SIA6 expression in colon cancer patients had no correlation with overall survival and relapse-free survival. The low expression of ST8SIA in colon cancer patients with high-grade colon cancer may be associated with poor outcomes after primary treatment. Thus, the relationship between the expression of ST8SIA6 and survival rates should be studied by collecting more data.

We further revealed the ST8SIA6 co-expression networks in colon cancer. According to GO function annotation results and KEGG pathway enrichment analysis, the upregulated functional genes with ST8SIA6 were mainly enriched in cell surface adhesion, angiogenesis, membrane vesicle trafficking, and lipid-mediated metabolic pathways. A recent study has found a network constructed to identify TDP2, PLOD2, FREM1, ARHGAP18, and FABP1 genes. Bian et al. [42] revealed that the regulation of TDP2 activity by post-translational modification is significantly increased by cancer cells’ DNA damage response and chemoresistance in patients. PLOD2 is a potential therapeutic target that increases cancer cell resistance and inhibits drug-induced apoptosis [43,44]. Li et al. [45] revealed that FREM1 is a favorable prognostic marker and is correlated with immune infiltrating levels in breast cancer. Overexpression and downregulation of ARHGAP18 inhibit metastasis by regulating cytoskeleton remodeling [46,47]. Liu et al. [48] demonstrated that decreased FABP1 is related to gastric disease progression by regulating tumor immunity. These networks are associated with ST8SIA6, which affects cancer cell metabolism, invasion, metastasis, tumor-associated microenvironment, and immune infiltration status. The findings from unpublished studies demonstrated that the downregulation of ST8SIA6 in COAD tissues from our Biobank promoted cancer stemness and migration in colon cancer by increasing membrane-bound protein levels. Thus, ST8SIA6 co-expressed genes and ST8SIA6 may drive multiple oncogenic pathways with abnormal glycosylation in cancer cells and serve as a potential tumor biomarker for colon cancer.

Finally, we predicted that alterations in the expression of ST8SIA6 in colon cancer are involved in tumor immunity and immunotherapy responses based on transcriptomic analysis. The clinical implications of glycosylation modification as a biomarker and molecular modifier of antibody-mediated immunity are explored in [49,50]. Targeted disialic acid is generated by ST8SIA6 and Siglec specifically in tumor or pathogen infection, and has already shown treatment efficacy [51,52]. However, disialylation may contribute to ST8SIA6 and ST8SIA3 expression. ST8SIA3 deficiency reduces the disialic acid level [15]. Recently, Friedman et al. [25] demonstrated that ST8SIA6 overexpression promoted tumorigenesis by interacting with disialic acid in cancer cells and Siglec-E in macrophages. Altered immune responses in the tumor microenvironment, which are dependent on Siglec-E or Siglec-7/9 expression, have been found in tumor-infiltrating cells from murine tumor models or human cancer [53,54]. Targeted interaction disialylation by ST8SIA6 on tumor cells and tumor-infiltrating immune cells is important for cancer immunotherapy. According to TIMER-provided immune cell infiltration in TCGA tumor profiles, cells of the innate immune system with ST8SIA6 might have a strong influence on the pathogenesis of colon cancer. On the TISMO website, different immune checkpoint molecule treatment response groups group analyses showed that the expression of ST8SIA6 with antiPD1 treatment might have a strong immune response in colon cancer.

## 5. Conclusions

In conclusion, this study provides multilevel relationships for the oncogenic roles of ST8SIA6 and its co-expressed genes in colon cancer and its potential as a diagnostic marker for colon cancer. Our results suggest that ST8SIA6 downregulation in colon cancer is related to multiple biological effects. Furthermore, ST8SIA6 has been linked to the tumor-associated microenvironment and immunotherapy responses. We also identified ST8SIA6 co-expression genes in colon cancer using more molecular biology experiments and further clinical research. These findings aim to help achieve personalized medicine using bioinformatics analysis for the diagnosis and treatment of colon cancer.

## Figures and Tables

**Figure 1 jpm-12-00401-f001:**
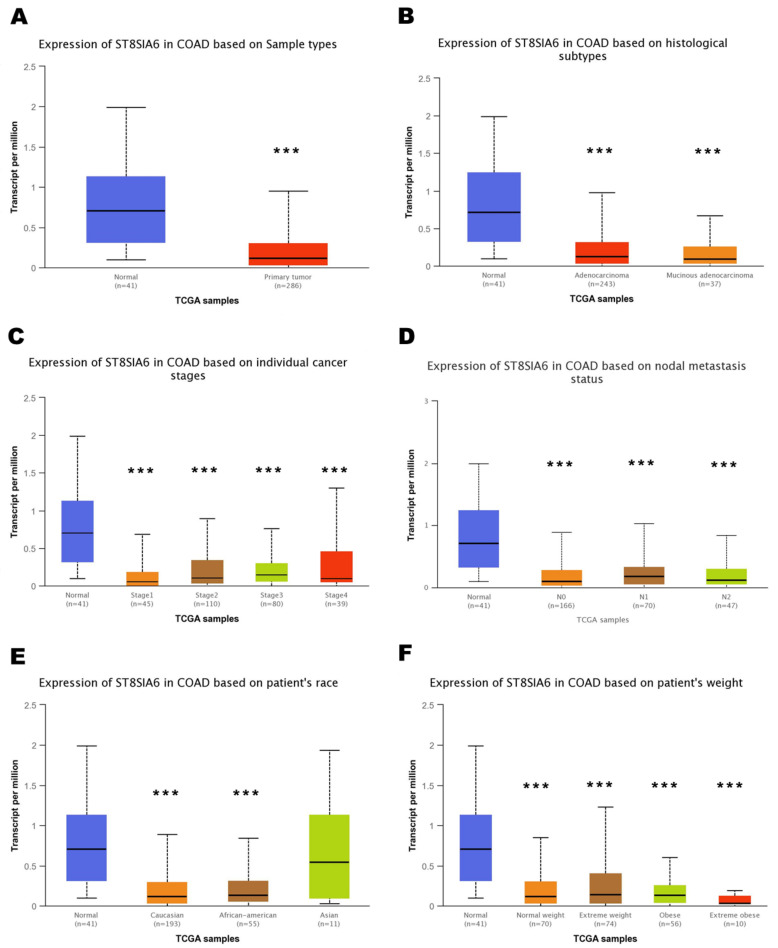
The mRNA expression of ST8SIA6 in colon cancer based on the TCGA database from UALCAN. (**A**) Boxplot showing relative expression of ST8SIA6 in normal individuals or colon adenocarcinoma (COAD) patients. (**B**) Boxplot showing relative expression of ST8SIA6 in normal individuals or COAD patients based on histological subtypes. (**C**) Boxplot showing relative expression of ST8SIA6 in normal individuals or COAD patients with 1, 2, 3, or 4 stages. (**D**) Boxplot showing relative expression of ST8SIA6 in normal individuals or COAD patients based on nodal metastasis status. (**E**) Boxplot showing relative expression of ST8SIA6 in normal individuals of any ethnic group or COAD patients of Caucasian, African-American, or Asian ethnic group. (**F**) Boxplot showing relative expression of ST8SIA6 in normal individuals or COAD patients with average weight, extreme weight, obesity, or extreme obesity. *p*-value Significant Codes: *** <0.001 compared with normal group.

**Figure 2 jpm-12-00401-f002:**
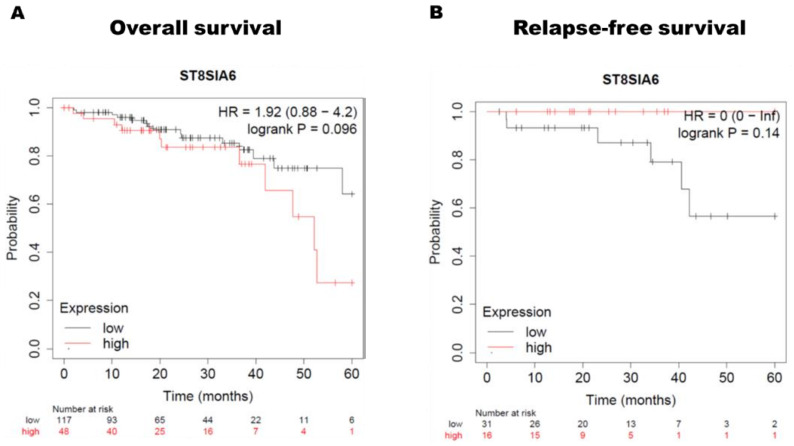
Survival curve analyses show the correlation between ST8SIA6 mRNA expression and (**A**) Overall survival (OS) or (**B**) Relapse-free survival (RFS) in patients with colon cancer. HR = hazard ratio. HR and 95% CI were estimated.

**Figure 3 jpm-12-00401-f003:**
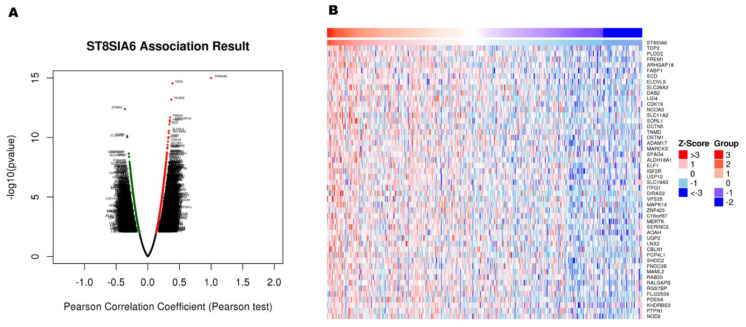
Using LinkedOmics to identify the relationship between genes differentially expressed and ST8SIA6 correlations in colon cancer. (**A**) Pearson correlation coefficient used to analyze correlations between ST8SIA6 and differentially expressed genes in colon cancer. (**B**) Heat maps representing the genes positively correlated with ST8SIA6 in colon cancer.

**Figure 4 jpm-12-00401-f004:**
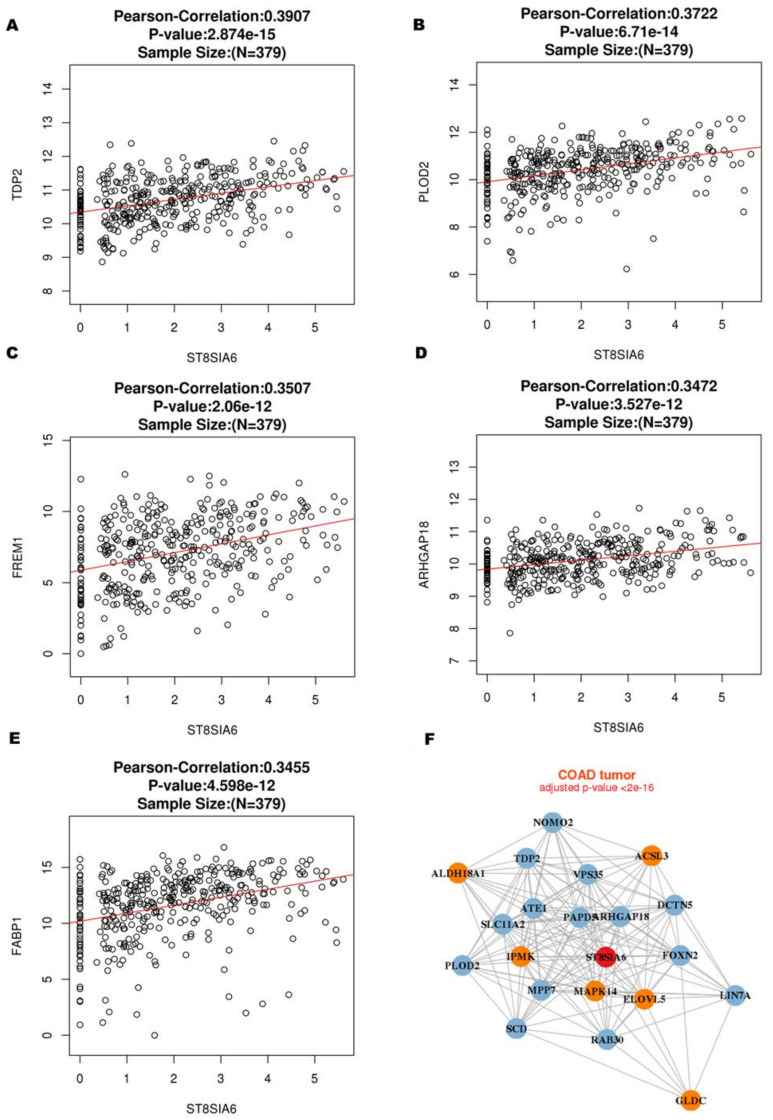
The positive genes correlated with ST8SIA6 involved a metabolic pathway in colon cancer. (**A**–**E**) Top 5 co-expressed genes with the highest correlation with ST8SIA6 in colon cancer. (**F**) Gene–Gene interaction network of ST8SIA6 and differentially co-expressed genes from ccmGDB. Red nodes represent input gene, oranges nodes represent cell metabolism genes.

**Figure 5 jpm-12-00401-f005:**
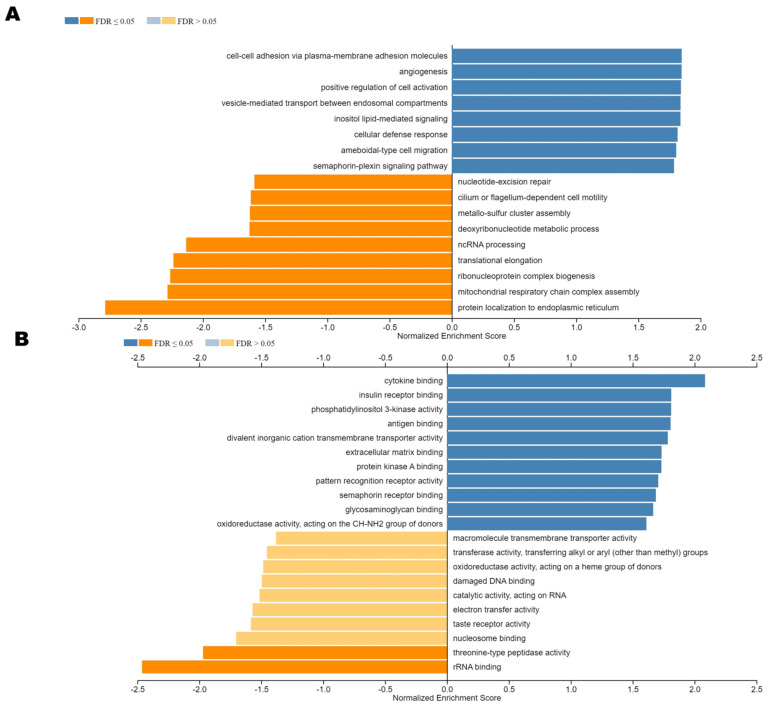
Enriched GO functions and KEGG pathways of ST8SIA6 in colon cancer. The identified differentially expressed genes with ST8SIA6 in colon cancer were explored from LinkedOmics for GO and KEGG pathway analyses. (**A**) GO term annotation of ST8SIA6 and co-expressed genes by gene set enrichment analysis (GSEA). (**B**–**D**) KEGG pathway-related Panther or Reactome analysis showed the enrichment pathway related ST8SIA6 and co-expressed genes.

**Figure 6 jpm-12-00401-f006:**
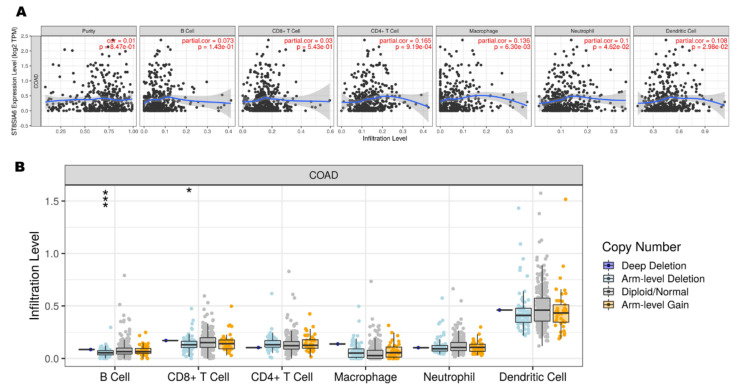
Correlations of ST8SIA6 expression with immune infiltration level in colon cancer. (**A**) ST8SIA6 expression is significantly related to tumor purity and has significant positive correlations with infiltrating levels of CD4+ T cells, macrophages, neutrophils, and dendritic cells in colon cancer. (**B**) ST8SIA6 CNA affects the infiltrating levels of B cells and CD8+ T cells in colon cancer. *p*-value Significant Codes: *** <0.001, * <0.05 compared with arm-level deletion, diploid/normal group, and arm-level gain.

**Figure 7 jpm-12-00401-f007:**
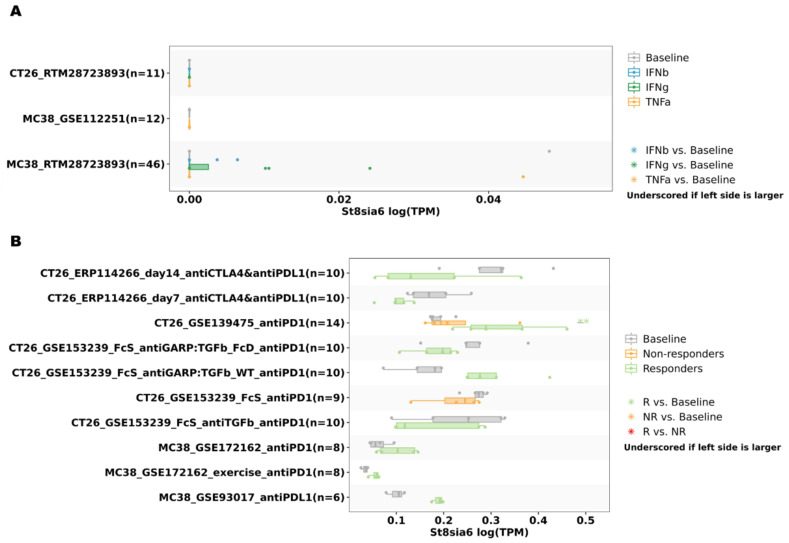
Correlations of ST8SIA6 expression with cytokine treatment and immunotherapy response in CT26 and MC38 colon cancer cells. (**A**) ST8SIA6 expression is not significantly related to interferon gamma, interferon beta, and TNF alpha treatment in the CT26 and MC38 colon cancer models. (**B**) ST8SIA6 expression in CT26 or MC38 colon cancer model is stimulated by different immune-checkpoint blockade (ICB) treatments. Boxplot showing ST8SIA6 expression before and after cytokine treatments and ICB treatments in different comparison conditions. *p*-value Significant Codes, ** <0.01, compared with baseline group.

**Figure 8 jpm-12-00401-f008:**
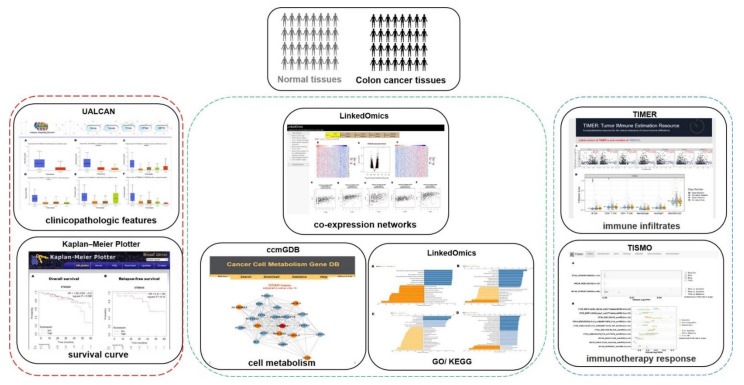
Workflow of the study. The graphical figure indicates a bioinformatics analysis of ST8SIA6 gene expression as tumor marker of colon cancer using web based and high-throughput approaches.

## Data Availability

The data that have been used to support the findings of this study can be requested through the corresponding author. The data sets can be accessed at http://ualcan.path.uab.edu/, http://kmplot.com/analysis, http://www.linkedomics.org/admin.php, http://bioinfo.mc.vanderbilt.edu/ccmGDB, http://timer.cistrome.org, and http://tismo.cistrome.org (accessed on 24 February 2022), respectively.

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
