# Peer review of "Bioinformatics Analyses Identify the Therapeutic Potential of ST8SIA6 for Colon Cancer"

_jpm, 2022, doi:10.3390/jpm12030401_

Round 1

Reviewer 1 Report

This study uses several gene databases to investigate the role of ST8 sialyltransferase in the formation of colon cancer. Overall it is well written and   uses multiple studies to tell a story. I have several specific comments

1) The abstract is not well written as opposed to the main paper. The English is worse and the concepts are not described well enough to understand what the paper will study. This should be overhauled.

2) I was confused about the Kaplan Meyer section. Overall survival is better with low expression but disease free survival is worse. These discordant results only serve to confuse the reader. In any case, P values are not significant and this was not well expressed. I would either remove this completely or state that nothing was significant and therefore nothing can be concluded. As it stands now, I feel the paper is trying to at best prove a point that does not exist, and at worst is trying to trick the reader.

3) The abbreviations are extensive and not well elucidated. For example, TCGA was never described as The Cancer Genome Atlas. Abbreviations should all be spelt out completely and simplified as much as possible.

Reviewer 2 Report

In this manuscript entitled « Bioinformatics analyses identify the therapeutic potential of 2 ST8SIA6 for colon cancer”, the authors Ko et al. identify ST8SIA6 as a therapeutic target for colon cancer.

Many flaws

-Figures are way to small and should enlarged

-Not sufficiently detailed context of the study

-References should be added

-Additional experiments should be carried out

The introductory part on α2,8-sialyltransferases (ST8SIA) and in particular on ST8SIA6 is not sufficient. Line 70: These glycoproteins in mammals that are di-sialylated should be presented (reference 13 does not concern these glycoproteins but glycolipids).

Line 71 human disease-related disialylation should be detailed as well. The human ST8SIA6 was cloned and its enzymatic specificity towards O-glycoproteins described long ago.

In general, almost all sialyltransferases are overexpressed in cancer and hypersialylation is observed. Curiously, the mRNA expression of ST8SIA6 is decreased in all the cancer tissues in the TCGA database. Is the underexpression of ST8SIA6 in COAD patient significant?

The recently published study showing that ST8SIA6 promotes tumor growth in mice should be mentioned and discussed.

Another concern is that it is difficult to understand if selected datasets are identical from one database to another

No clear conclusion can be drawn from each bioinformatics analysis: Line 166 data obtained are not conclusive and Line 193 part of the sentence is missing.

Line 237 We need more details on CT26 and MC38 colon cells and what they are used for.

Line 179 and 277 What are the relationships between ST8SIA6 and these other genesPLOD, TDP2…?

Line 288, It is not clear what the ST8SIA6 related genes are as ST8SIA6 is unique in the human genome.

ST8SIA6 and not ST8SAI6

Round 2

Reviewer 1 Report

Corrections that were made have improved the quality of the paper and it reads better. I have no concerns.

Reviewer 2 Report

In the abstract, ST3- and ST6-sialyltransferases should be noted α2,3- and α2,6-sialyltransferases which refer to all the ST3GAL,ST6GALNAC and ST6GAL. In addition, α2,8-sialyltransferases (ST8SIA) which include ST8SIA6 should be mentioned.

References to di-Sia proteins have been added in the introduction, but the relationships of one or more α2,8-sialyltransferases and di-sialylation of glycoproteins are still not given. Reference should be made to the first studies that reported synthesis of the tri-sialylated core 1 structure (Neu5Acα2,8Neu5Acα2,3Gal β1,3 [Neu5Acα2,6] GalNAc) on glycoproteins mediated by the human ST8SIA6 (Teintenier-Lelievre et al. Biochem. J. 2005) and by the mouse enzyme (Takashima et al. JBC, 2002). In addition, a short description of the other ST8SIA that could be involved (ST8SIA3, ST8SIA2 and ST8SIA4) with ad hoc references should be made. Finally, as correlation between di-Sia glycans and ST8SIA are not well established it should be mentioned in the text that enzymes other than ST8SIA6 might contribute to the synthesis of these di-sialylated structures found on glycoproteins and modulation of their expression level.

To the issue #7, the authors answered that ST8SIA6 gene expression is decreased in all cancer tissues in the TCGA database and that this is not the case in the GEPIA dataset. However, lower expression of ST8SIA6 in COAD tumor compared to normal tissue is observed in the two datasets. This should be mentioned in the discussion and data not necessarily shown as it reinforce the idea of potential target for colon cancer.

Investigations were conducted to know the role of ST8SIA6 in colon cancer (Line 149 and Line 204). It is not clear to me what the roles (biological functions) of ST8SIA6 in colon cancer are exactly. Is there any modification of the levels of di-Sia motifs in colon cancer that could be demonstrated? What are the conclusions relative to ST8SIA6 being a potential therapeutic target for colon cancer? This should be discussed also in the text

A lot of sentences are still not clear and should be rephrased:

Line 219, Line 221, Line 226, Line 290, Line 328, Line 334

“ST8SIA6 related genes” should be changed to ST8SIA6 co-expressed genes.
